# Model-Driven Developed Terminal for Remote Control of Charging Station for Electric Vehicles Powered by Renewable Energy

Jovan Vujasinović *, Goran Savić and Milan Prokin

School of Electrical Engineering, University of Belgrade, 11000 Belgrade, Serbia
* Correspondence: jovan.vujasinovic@vfholding.rs

**Abstract:** A terminal for remote control of charging stations for electric vehicles (EV) powered by renewable energy has been presented in this paper. This terminal enables remote control of EV chargers, smart batteries, smart electricity meters, fiscal cash registers (FCR), as well as remote control of renewable energy sources and other devices within the station. This terminal also makes charging stations more accessible to electric vehicles users, to electricity distribution companies, to electricity suppliers, to tax administrations, and finally to users and owners of charging stations. Therefore, communication and control with all these devices and systems is integrated in one device. Realization of hardware and software of such a terminal has also been described in this paper. The net result of development and commercialization of terminals would encourage an increase in the use of electric vehicles powered by energy from renewable sources, which would cause a decrease in the level of air pollution and all negative effects it causes in the future. Different categories of this device are considered. Moreover, although it is a device with embedded software, a very advanced method was used, that is, a model-driven development method, which enables fast and more efficient development and maintenance of the device. The results of the application of this method to the terminal for remote control of fiscal cash registers are provided. They were compared with the results of the development of the terminal for remote control of smart meters without applying this method. A simulation of the development of the terminal for remote control of the station is also provided. The presented method can be used in the future for faster and better-quality development of embedded software.

**Keywords:** electric vehicles chargers; model-driven development; remote control; renewable energy sources





## 1. Introduction

Production and use of electric vehicles are constantly being increased. The benefits of that trend in terms of air pollution reduction can be fully achieved only if electric vehicles are powered by energy obtained from renewable sources. Further development of the infrastructure for EV charging becomes more and more important, especially in terms of increasing the number of charging stations for electric vehicles powered by renewable energy sources [1].

Charging stations for electric vehicles powered by renewable energy are being integrated into one larger system, which increases their accessibility to electric vehicles users, electricity distribution companies, suppliers, and tax administrations, as well as to the owners and users of the EV charging stations. Such integration increases the efficiency of using the electricity distribution network and saves time and money. One of the most important features of such a system is remote control of the station for EV charging. Remote control enables the control of the mentioned devices from one center. For example, the control of all smart meters from one center allows the distribution operator to manage the power grid on the consumption side, which became a necessity due to the emergence of

many small- and medium-sized sources of electricity, as well as electric vehicles. Control of all smart batteries from one center allows the supplier to optimize electricity trading and achieve higher returns. The control of all fiscal cash registers from one center enables the tax administration to easily and quickly monitor the collected tax and reduce tax evasion. Control of all EV chargers from one center allows electric vehicle owners to have timely information on the optimal place for them to charge their vehicles. In addition to this, remote control allows station owners to manage it in real time wherever they are. The key device in this system is a terminal for remote control of charging stations for electric vehicles powered by renewable energy. The realization of this terminal is presented in this paper.

Technical requirements for this terminal are to enable remote reading all registers, remote setting all the parameters, and starting all possible actions in EV chargers, smart batteries, smart meters, and fiscal cash registers by electric vehicles users, electricity distribution operators, suppliers, and tax administration, as well as for the owners and users of the EV charging stations. For smart meters, there are the registers of active energy and reactive energy, the billing profile, the maximum demand and the quality of electricity, event logs, etc., the parameters of tariff policies, power limitation, etc., and actions of connecting or disconnecting consumers, etc. For smart batteries, there are registers of capacity status, billing profile, etc., parameters of operating mode (charging/discharging), permission (permission/prohibition of charging), energy price, etc., actions of charging battery from the grid, discharging battery to the grid, charging battery from the renewables, and transferring energy from the renewables to the grid. For EV chargers, there are registers of total consumed energy, daily consumed energy, billing profile, etc., the parameters of limiting the maximum consumption during the day, energy price, operating mode (charging/discharging), and permission (permission/prohibition of charging), etc. For fiscal cash registers, there are registers of turnovers, resets, etc., and parameters of tax rates, articles, prices, and cashiers, etc. In addition to this, advanced terminal functionality is an automatic mode of operation where the terminal makes decisions about what to do with the battery, charger, and renewable sources of energy depending on the variable inputs (current energy price, current energy consumption, current energy production, etc.).

The system's end-to-end security architecture must offer an ICT foundation for data protection and functionality across all communication network segments, from EV chargers, smart batteries, smart meters, and fiscal cash registers to electric vehicles users, electricity distribution operators, suppliers, and tax administration, as well as to the owners and users of the EV charging stations. Although there are many particulars to system implementation, the implementation of security infrastructure is very similar to that of other highly secure ICT infrastructures (such as banking, insurance, government, etc.), and it is necessary to use industry standard and tested security elements and solutions. On the basis of radio frequency technologies, terminals create a mesh network. Before being provided access to the information infrastructure, each terminal that joins the mesh network must first prove its identity. Extensible Authentication Protocol (EAP), RADIUS, and IEEE 802.1x are just a few examples of open protocols that can be used to their fullest potential for strong node authentication. Link-layer encryption must be used in the mesh architecture. The terminals must implement an application security layer that, at a minimum, meets suite #0 of the DLMS COSEM standard (AES GCM 128). A cryptographic key management system is required for terminals that contain cryptographic keys for authentication, encryption, integrity, or other cryptographic operations. This system must offer sufficient key diversity and appropriate protection for cryptographic materials.

Recently, we have witnessed the phenomenon that many households practically become charging stations for electric vehicles powered by renewable energy. This happens for two reasons. The first reason is that, in addition to switching to electric cars, owners install chargers in their homes. Another reason is that more and more households are becoming prosumers, that is, both buyers and producers of electricity at the same time. Every consumer can install solar panels or wind generators at home in order not only to

produce and use the produced electricity for their own needs but also, in the case of surplus, to have the right to enter the market, sell the electricity, and receive compensation for it. This enables the consumer to reduce his costs and increase his income by his own initiative. Consumers also have the possibility to join together or establish local energy communities, which would bring together more consumers and thus, at the same time, meet their own needs for electricity and be more competitive in the market when selling excess production. Aggregators are market participants who combine consumption or production electricity from several prosumers and trade it on the market.

The popularity of electric cars has been rising over time. In some countries, the number of electric vehicles has been rising year over year. However, the market for electric vehicles in some other countries is still very nascent, as evidenced by the lack of infrastructure for vehicle charging or the insignificant demand for such services in locations where such infrastructure has already been established, not to mention the meager sales of electric vehicles [2]. Therefore, the development of infrastructure and services for vehicle charging and integration of renewable energy sources into the system is of great importance for further development of the electric vehicles industry.

Fuel cells play a significant role in the drive application of automobiles due to their benefits of high efficiency, high power density at low temperature, quick startup, and zero pollution [3]. The fuel cell is a cutting-edge energy device with a wide range of uses. The fuel cell vehicle stands out among them for its benefits to the environment, zero pollution, and great efficiency.

Due to an increase in the number of electric cars, the prices of materials necessary for the production of batteries has been increased several times in recent years. That is the reason why recycling batteries is becoming increasingly important and very popular since not every cell in a used battery has already met the requirements for its end of life and can still be used for subsequent purposes.

This paper has the following structure. The related work is presented in Section 2. In Section 3, the architecture of the system for remote control of charging stations for electric vehicles powered by renewable energy has been described. Section 4 contains a description of subsystems of the terminal for remote control of the charging station for electric vehicles. Hardware of the terminal for remote control of the charging station for electric vehicles has been described in Section 5. Different variants of terminal hardware implementation are presented in Section 6. Section 7 contains a description of the software of the terminal for remote control of EV charging stations. In Section 8, drivers for the terminal have been described, while the subprograms for executing the processes are disclosed in Section 9. The discussion and results are presented in Section 10, and conclusions have been provided in Section 11.

## 2. Related Work

Papers dealing with individual subsystems can be found. Thus, papers [4,5] show the analysis of terminal hardware and software for the system for remote control of meters, while papers [6,7] show the analysis of hardware and software for the system for remote control of fiscal cash registers, while papers [8–13] provide an analysis of a system for remote control of EV chargers. Paper [14] provides an overview of works dealing with systems and terminals for remote control of charging stations for electric vehicles powered by solar power plants. Reference [15] shows a system for remote control of smart batteries.

One can find works that deal with model-driven development, that is, object-oriented way of programming embedded software, although they are not common. Further, the application in practice is still not common due to less control of the programmer over the speed of execution and the amount of program and working memory required for the operation of the software obtained in this way, which is very important for embedded software. Paper [16] provides a model for object-oriented programming that is applied in the automotive industry. Paper [17] discusses the trend that embedded software development is shifting from manual programming to model-driven development (MDD) and why it

is important to assure the quality of embedded software. Work [18] elaborates verifying protocol conformance using software model checking for the model-driven development of embedded systems.

Compared to the abbreviated version [19], this paper is much more detailed (i.e., contains a general definition of the model, i.e., the process) and provides an overall overview of the hardware and software and in particular adds a detailed explanation of the procedures for executing the process using model-driven development, that is, the object-oriented way programming.

There are many recent papers demonstrating high scientific interest in solving the aforementioned problems, just to name a few. A voltage balancer can be adopted to suppress the unbalanced current for each node of the neutral line, leading to its unbalanced voltage in a bipolar DC distribution network, caused by the unbalanced load resistance, line resistance, and renewable energy source [20]. Using the internal converter of a doubly fed induction generator (DFIG)-based wind turbine to provide voltage support auxiliary service is an effective scheme to suppress the voltage fluctuation at the point of common coupling (PCC) [21]. A local energy community with different types of prosumers is optimized (household, commercial, and industrial), and each of them is equipped with a photovoltaic panel and a battery system [22]. The financial and economic benefits related to EV management in Vehicle-to-Building (V2B), Vehicle-to-Home (V2H), and Vehicle-to-Grid (V2G) technologies are presented in [23]. Renewable energy sources and EV growth provide new challenges for grid stabilization, requiring smart grid techniques to reconfigure and compensate for load fluctuation and stabilize power losses and voltage fluctuation [24]. The growing penetration of electric vehicles can pose several challenges for power systems, especially distribution systems, due to the introduction of significant uncertain load, which can be solved by clustering methods [25]. According to the randomness of photovoltaic power generation and EV charging, the dynamic response capability, power support capability, effective convergence time, system stability, system failure rate, and other characteristics of regional loads are comprehensively analyzed, and the grid energy management model of EV charging network and distributed photovoltaic is proposed, while, according to certain statistical characteristics, the distributed photovoltaic will be concentrated, and EV charging will be prioritized to achieve nearby consumption [26]. The ZED 2i depth sensor can be utilized in a robot-based automatic EV charging application for further enhancement [27].

## 3. System Architecture

In Figure 1, the block diagram of the architecture of the system for remote control of a charging station for electric vehicles powered by renewable energy [19] is shown. The terminal for remote control of charging station for electric vehicles powered by renewable energy is the main component of the system. By Internet of Things (IoT) network, it is connected with renewable energy sources, smart battery, EV chargers, fiscal cash registers, smart electricity meter, user gadgets, and other gadgets. This terminal is also connected to the cloud via the internet. That fact enables many functionalities, such as monitoring, processing, setting, and storage of data received from renewable energy sources, smart battery for energy storage, EV chargers, smart meter, and fiscal cash registers. The access to the mentioned data in the cloud is possible through several different control centers: EV chargers control center, supplier control center, electricity distribution company control center, and tax administration control center. The owners of electric vehicles receive all information about EV chargers through the EV chargers control center, while the electricity available in the system is traded through the supplier control center. The owners of EV charging stations also have access to the data stored in the cloud. In order to process these data, advanced algorithms have been used, which increases the efficiency of distribution network use, as well as allows significant savings in the system and enables the implementation of innovative smart energy services.

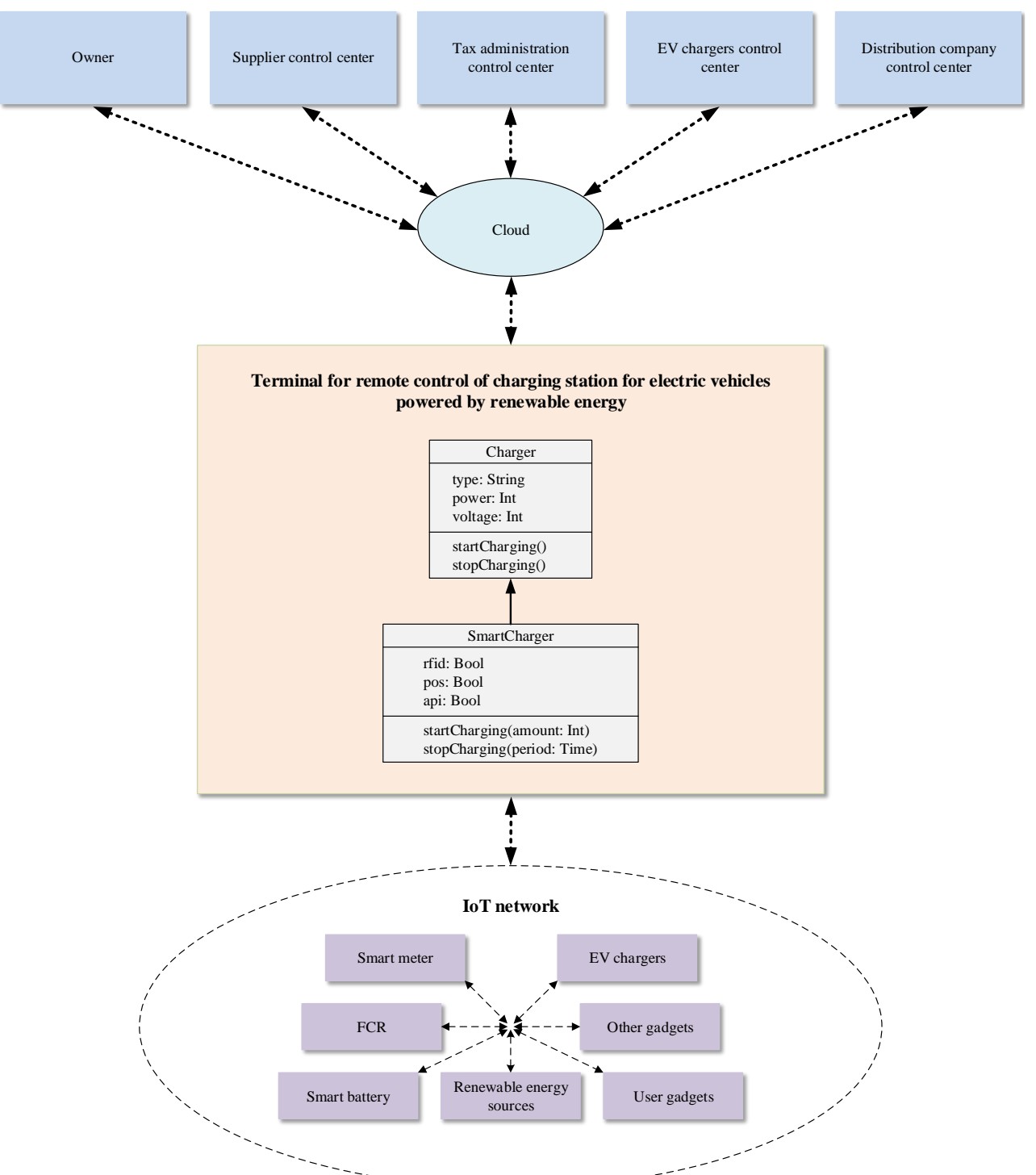

**Figure 1.** Block schematic of the design of the system for remote control of charging station for electric vehicles powered by renewable energy.

## 4. Description of Subsystems

Four subsystems are included in the system for remote control of renewable energy sources powered station for electric vehicles (Evs) charging: a subsystem for remote control of EV chargers, a subsystem for remote control of smart batteries, a subsystem for remote control of smart meters, and an optional subsystem for remote control of fiscal cash registers.

### 4.1. The Subsystem for Remote Control of EV Chargers

EV chargers, a terminal for remote control of EV chargers, and an EV chargers control center make up the three main parts of the subsystem for remote control of EV chargers [14]. In Figure 2, this subsystem is depicted. Energy from smart batteries is used to power electric vehicles. On the other hand, smart batteries receive their energy from renewable sources or, if necessary, the electricity distribution network.

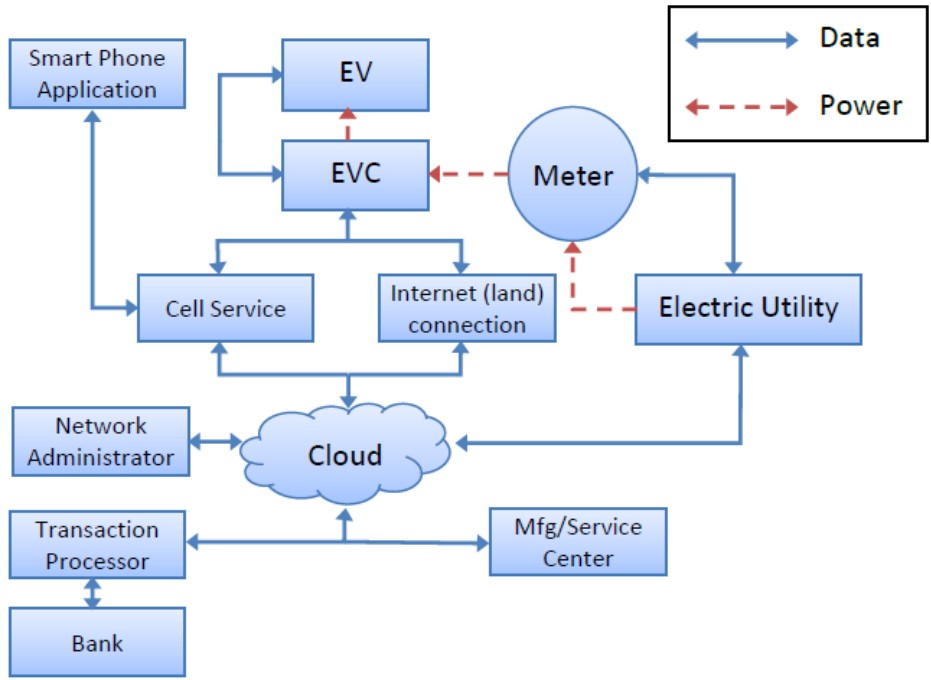

**Figure 2.** The subsystem for remote control of EV chargers.

The terminal for remote control of EV chargers has access to information from the chargers, such as power availability, charging programs, and prices. These data are transmitted over the internet to the EV chargers control center, where they are then made accessible to end users. The pricing of power at EV charging stations is managed by this subsystem using software algorithms based on the stations' locations and the amount of energy that is available. With this strategy, it is possible to lower the price of electricity at stations with more power available (to encourage customers to charge their electric vehicles at these stations), while the price of electricity may be raised at stations with less power available (so that customers would be discouraged from supplying their electrical vehicles with electricity at these stations). The owner of the charging station, if it is a separate subsystem or artificial intelligence technologies operating in a unified terminal with access to all devices (EV chargers, smart batteries, smart meters, and fiscal cash registers), appropriate control centers, information systems, and software platforms, decides on price adjustments.

### 4.2. The Subsystem for Remote Control of Smart Batteries

Smart batteries for energy storage, terminals for remote control of smart batteries, and a supplier control center make up the three major elements of the subsystem for remote control of smart batteries [15]. In Figure 3, this subsystem is depicted. The smart battery consists of an inverter and a battery. Smart battery management involves optimizing the operation of the inverter and the associated batteries [28]. Smart batteries for storing electricity obtain energy from the electrical distribution network or from renewable sources. The supplier control center receives information about the smart batteries' available energy from the terminal for remote control of smart batteries. This enables owners of EV charging

stations who are experiencing a temporary energy shortfall to purchase energy (obtained from renewable sources) from producers who are experiencing a temporary energy surplus. Similar to this, proprietors of charging stations for electric vehicles who currently have an excess of accessible energy can sell that energy to consumers who currently have a shortfall of available energy. Due to increased renewable energy production at a given time, this also enables owners of EV charging stations to purchase excess energy at reasonable prices and store it in smart batteries for energy storage, where it can later be sold at higher prices when there is an energy shortage in the system (due to reduced production of energy from renewable sources at a particular time). After purchasing electricity derived from renewable sources, the terminal for remote control of smart batteries is in charge of managing the smart battery charging process.

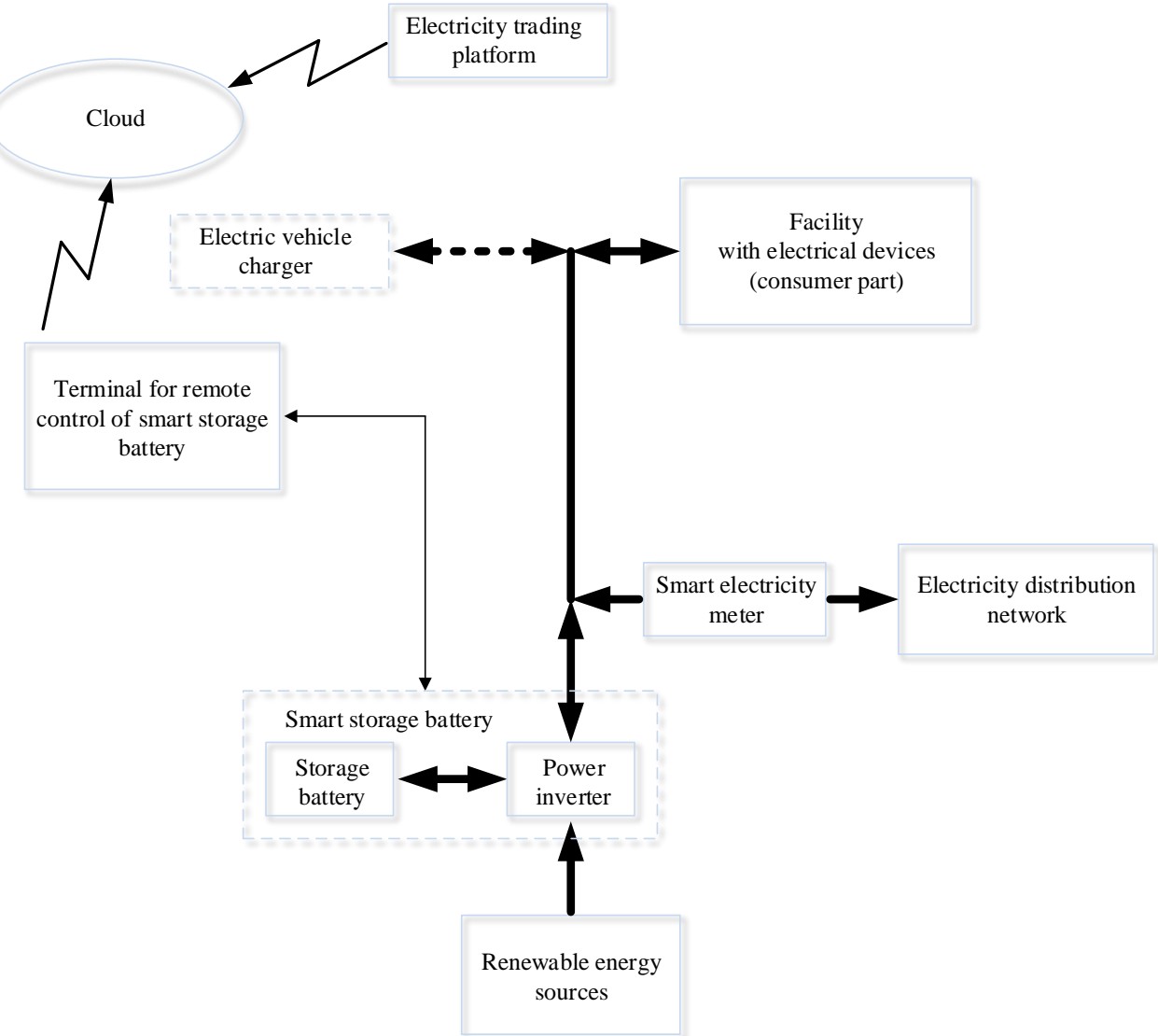

**Figure 3.** The subsystem for remote control of smart batteries.

Considering that there are batteries in the electric vehicles themselves, the use of those batteries (V2G concept) in vehicles has recently been considered a great deal. Since that concept depends significantly on the availability of the vehicles, in this work, a special independent battery—smart battery—is provided for the sake of reliability. However, certainly, a detailed analysis of whether and in which case vehicle batteries would be sufficient can be the subject of further work.

### 4.3. The Subsystem for Remote Control of Smart Meters

Smart meters, a remote control terminal for smart meters, and the distribution company control center make up the three basic parts of the subsystem for remote control of smart meters [4].

In Figure 4, this subsystem is depicted. There are numerous features in smart meters. Smart meters measure active energy and reactive energy, they register the average maximum power within a predefined period, they measure the quality of electricity, and show the appropriate data on the display. Smart meters also offer flexible tariff policies and maintain metering integrity. Smart meters can limit the amount of consumed power, record event logs, and connect or disconnect consumers from the power grid remotely. They can also record profiles of the corresponding measured quantities. In order to gather data from meters, configure meters, set settings, and manage consumption, smart meters and the terminal for remote control of smart meters communicate with each other. On the other hand, the terminal for remote control of smart meters interacts with the electricity distribution company control center. This control center gathers data, manages smart meter settings, and generates reports, in addition to carrying out the administration of components of the subsystem for remote control of smart meters.

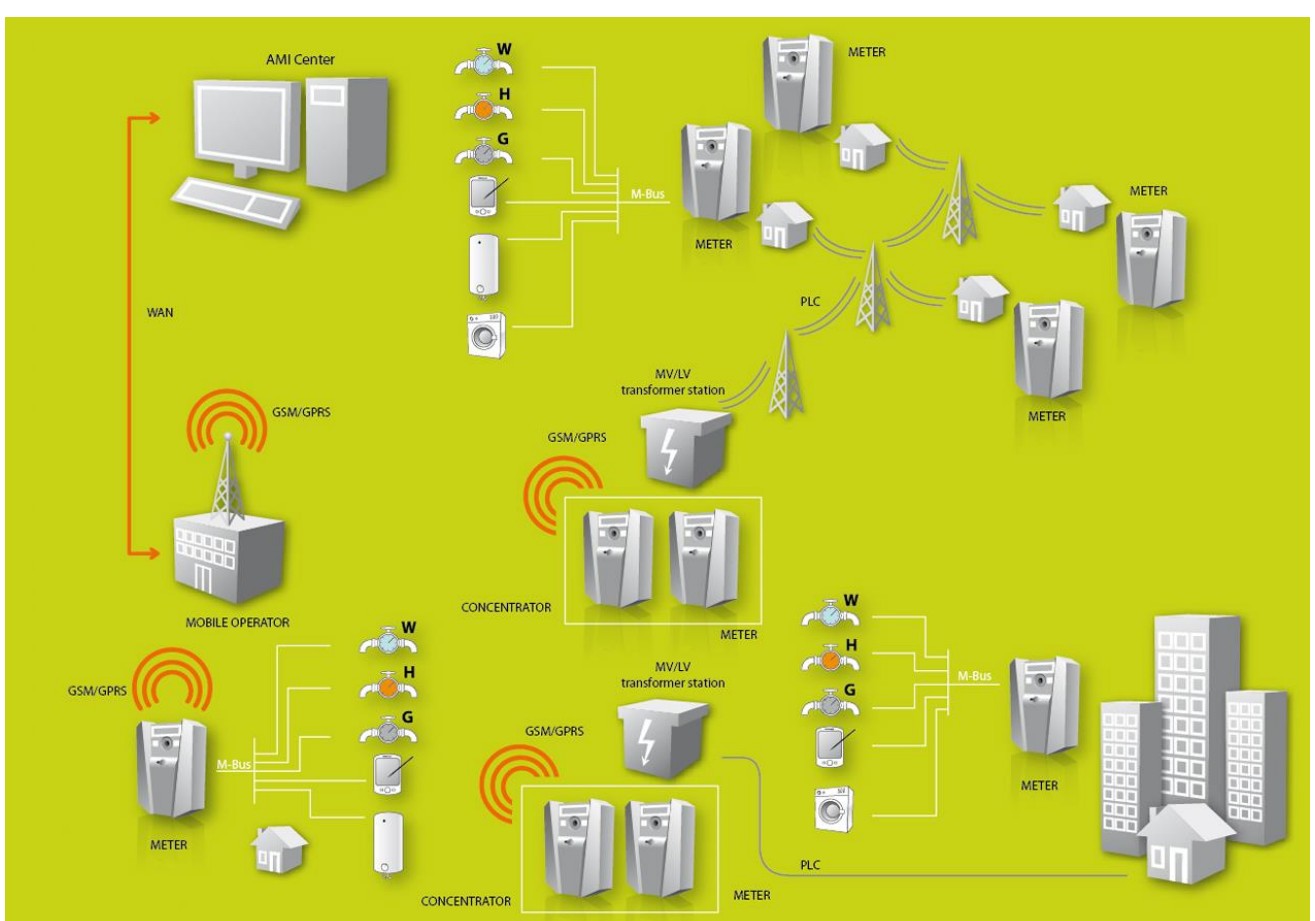

**Figure 4.** The subsystem for remote control of smart meters.

### 4.4. The Subsystem for Remote Control of Fiscal Cash Registers

Fiscal cash registers, terminals for remote management of fiscal cash registers, and the tax administration control center make up the three main parts of the subsystem for remote control of fiscal cash registers [6]. In Figure 5, this subsystem is depicted. Fiscal cash registers have a keyboard that allows the cashier to enter commands and to monitor the data on recorded transactions. Additionally, they have the ability to print the data on the

fiscal account and save the data in operational memory and fiscal memory. The data on the realized recorded turnover and the realized refunded turnover can be grouped, summed up, and presented by tax rates, articles, and cashiers in fiscal cash registers. They can also use the appropriate input–output port to download all relevant data in electronic form. They also meet the essential security requirements. The terminal for remote control of fiscal cash registers enables fiscal cash register remote reading, management, and programming with information on article structures and pricing. Additionally, it downloads information from the fiscal cash registers and creates the necessary reports, which are subsequently sent to the tax administration control center. The terminal for remote control of fiscal cash registers sends reports on transactions at tax rates for a certain period, information on resets, and tax rate parameters to the tax administration control center. To ensure tax collection, the tax administration is in charge of keeping track of every transaction through its control center.

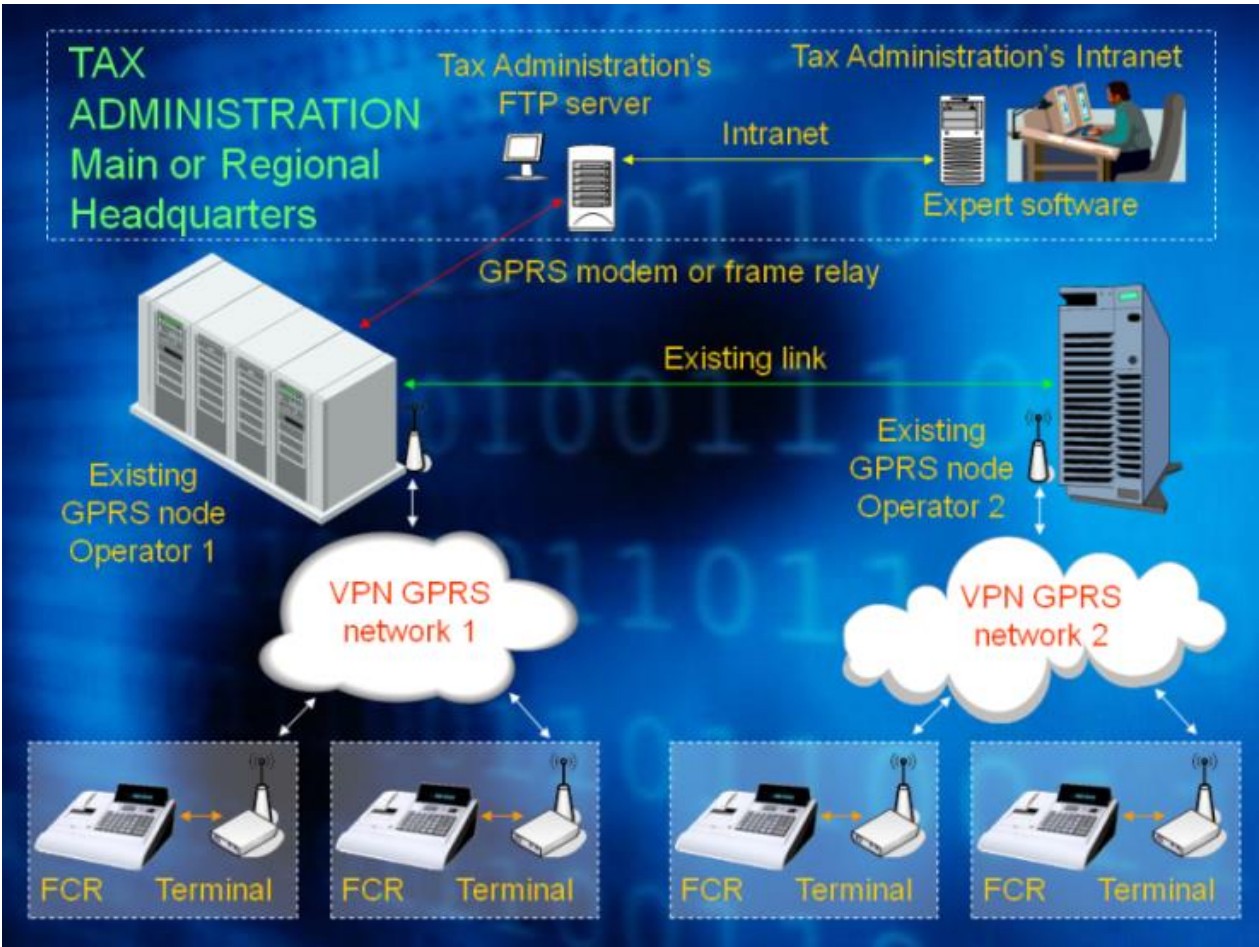

**Figure 5.** The subsystem for remote control of fiscal cash registers.

## 5. Hardware of the Terminal

The earliest terminal realizations for reading fiscal cash registers and printers are shown in [6]. The Ref. [7] feature advanced fiscal cash registers with integrated terminals with and without additional services. Electric car charger terminals for wireless control are displayed in [8].

Figure 6 displays a block schematic of the hardware for the terminal for remote control of the station for charging electric vehicles that is powered by renewable energy sources [19]. A microcontroller (µC) serves as the terminal's primary component and manages all operations. A microcontroller reset circuit must be included in the terminal hardware in order for the device to operate reliably. This circuit enables the microcontroller

to start up correctly after being powered on. For connecting to a personal computer for in-circuit programming and debugging purposes, this type of device often requires an appropriate connector that can only be accessed by opening the device enclosure. Additionally, if the microcontroller lacks a dedicated port for this use, additional connectors are required, to which the serial port of the microcontroller will be connected either to the proper terminal port (in this case, the cloud port) or to the connector previously mentioned in order to program the microcontroller by setting the appropriate jumpers. The first position is used when the terminal is in regular operation, while the second position is used to program and debug the terminal when it is being produced and serviced. The proper voltage regulators, together with supporting capacitors and resistors, are included in the power regulator. The input voltage is adjusted by these voltage regulators to all necessary voltage levels, which are then used to power every component of the terminal. In the event that the terminal is supplied by AC voltage, the power regulator additionally includes a corresponding rectifier of the AC voltage to DC voltage. Regardless of power disruptions, the real-time clock (RTC) offers correct time information. This integrated circuit ought to include a battery backup because of this. Non-volatile memory is used to hold the parameters required for the microcontroller software to function properly. These parameters must be saved permanently so that the terminal can continue to function even in the event of a power interruption. These purposes can involve use of EEPROM or FRAM memory. The terminal needs to store these data somewhere in the interim since data exchange with the IoT infrastructure and data exchange with the cloud occur at different times. It is anticipated that the internal operational memory of the microcontroller will not be large enough to store all these data because of its size. This is among the causes for the necessity of using more operating memory (SRAM) in the terminal hardware. Data in this memory should be maintained regardless of potential external power failures, which is another consideration. SRAM memory with battery backup of the power supply was previously employed (using the proper circuit with diodes or switches). FRAM memory is now employed for these applications.

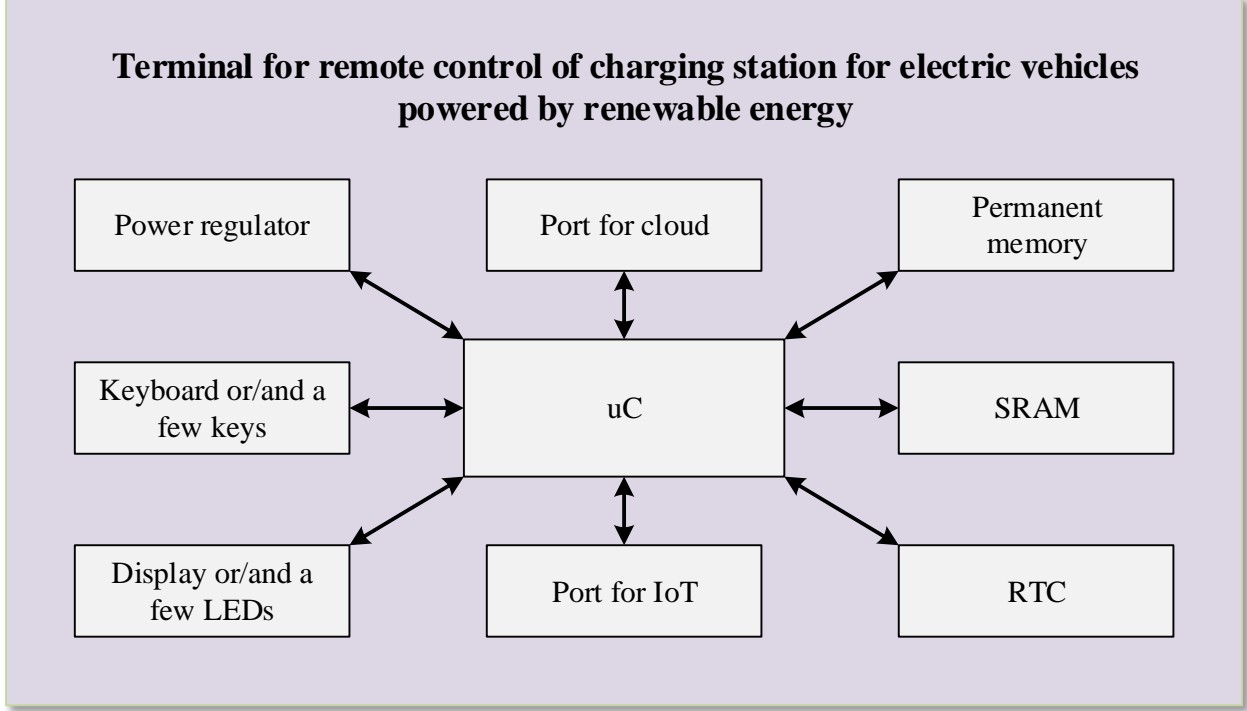

**Figure 6.** Block schematic of the hardware for the terminal.

A minimum of two ports are available on the terminal, one of which is used to communicate with the cloud and the other to the IoT network infrastructure [29]. As a result, a microcontroller should have at least two serial ports. Eventually, a microcontroller with only one serial port may be used, and the other serial port may be implemented using a separate integrated circuit. The microcontroller should have five serial ports in order to build a more advanced terminal. For serial connection with the suitable 5G/4G/3G/GPRS/GSM modem and the ETHERNET port, there would be two serial ports used for the cloud. If the microcontroller already has an internal implementation of an ETHERNET port, it is possible to realize directly; otherwise, it can be realized indirectly by using the serial port of the microcontroller and an external integrated circuit. The microcontroller's remaining three serial ports would be used to communicate with the IoT network infrastructure: one would be used to wire the station's devices, another to wirelessly connect those same devices, and the third would be used to wirelessly connect the station's customers (the port for local access is primarily for electric vehicle owners who use the station to charge their vehicles, but it can also be used by a repairman and station owners/operators). The wired connection of the devices within the station should be completed using an RS485 port, directly if the microcontroller already has one internal port built in or indirectly through the use of an extra integrated circuit. Zigbee or LoRa can be used to wirelessly connect station components, whereas WIFI is the ideal option for wirelessly connecting station users. An appropriate extra integrated circuit attached to the microcontroller's serial port is required to realize each of these interfaces.

It is typically necessary to provide a component for visual indication and a part with keys on the terminal depending on the functional requirements. In general, these two components provide direct communication between the operator or a repairman and the terminal. The section involving the keys can typically be implemented by employing a few special keys or the suitable keyboard. The operator or a repairman can input data or commands to the terminal or microcontroller through the component with the keys. Light-emitting diodes with accompanying resistors and inverters (in the basic variation) or displays (in the more advanced variant) can be used to implement the portion for visual indication. While the display is directly controlled by the microcontroller, the light-emitting diodes (LEDs) can be controlled by the microcontroller or by some peripherals, such as a 5G/4G/3G/GPRS/GSM modem (where LEDs indicate the presence of a 5G/4G/3G/GPRS/GSM network). The microcontroller displays all relevant information to the operator or a repairman using the component as a visual indicator.

## 6. Variants of Terminal Hardware Implementation

Depending on the target price and how sophisticated the functional needs are, different terminal hardware implementation options may be available. For the mentioned EV charging stations and households, there is a clear need for a device (terminal) that would combine these features. However, compared to industrial or commercial EV charging stations, households have distinct requirements and capabilities; thus, we suggest the three terminal classes of light, standard, and extended. For household use is intended the light class—for domestic charging stations. The light terminal should make it simple to configure, manage, and keep an eye on the right home gadgets. First and foremost, one or two EV chargers are anticipated to be connected to the light terminal, along with smart meters and smart electrical devices (smart water heaters, air conditioners, etc.) whose management can achieve economic management of electricity consumption in order to reduce costs, as well as the inclusion of users in the concept of a smart city (aspect of energy efficiency in the concept of a smart city) through communication with the smart city infrastructure. A renewable energy source, such as solar panels on the roof of the house [28], should also be able to be connected to the light terminal. The extended class is meant to meet the demands of industrial charging stations for electric vehicles with vast capacity, while the standard class is designed to meet the needs of commercial charging stations for electric vehicles that are the most basic. The extended class differs from the standard class only in that it

supports a far greater number of chargers and handles noticeably more data. Otherwise, it is substantially the same. It is used in huge parking lots where many of the parking places have chargers in them. These parking lots will be found in future large shopping centers, auto manufacturers, etc. For commercial and industrial charging stations, it is to be expected that the solar panels on the roof are not enough to power them. Whether wind, biomass or fuel cell, or some other renewable source can be used depends on the location of the stations and must be determined for each one individually. This analysis can certainly be the subject of further work.

In view of the best price–performance ratio, it is obvious that the light terminal can be realized with the smallest budget, whereas the extended terminal can be realized with the highest budget. A corresponding 8051 microcontroller with two serial ports (or perhaps one serial port and an additional integrated circuit for implementation of another serial port), EEPROM memory, SRAM memory with battery backup, a few buttons, and a few LEDs could, therefore, be used to implement the light terminal. The standard terminal can be implemented using an ARM microprocessor with five serial ports, FRAM memory, a corresponding keyboard, and a corresponding display. An industrial single-board computer with five serial connections, a great deal of memory, and ports for a keyboard and a screen can be used to develop the extended terminal.

Low price is very important for light terminals, while, for standard and extended terminals, it is not such a significant factor. The proposed hardware configuration for the light terminal already ensures the lowest possible price. However, what additionally affects the price reduction and what must, therefore, be considered is interoperability. Therefore, it is important that standardized communication protocols are used throughout the system as this also significantly reduces the price. This was also shown by the experience from the implementation of the smart metering system [30]. Even 20 years ago, many cost benefit analyses showed the technical and economic justification of installing this system in households as well. In the meantime, many distributions have started this process of mass replacement of smart meters. Therefore, there is already interest and economic justification for the installation of terminals for remote control of smart meters. Electric distribution companies, as well as suppliers, are interested in expanding the functionality of such a terminal to remote control of smart batteries and EV chargers. Since such an extension of the functionality of the terminal does not significantly affect its price, the economic justification of its application in practice is clear.

## 7. Software of the Terminal

Structured programming is typically used to develop software that is run by the terminal's microcontroller. The main program's algorithm must be defined, followed by the algorithms for each subprogram and the description of the memory layout for data storage, in order for the software to be implemented. The specifics of these algorithms rely on the terminal's technical requirements.

A series of subprograms, depicted in Figure 7, are executed at the start of the main program in order to start the terminal. Afterwards, the program goes into an infinite loop, where it checks the appropriate flags and calls the execution of the corresponding subprograms whenever the flags are activated (set). These programs can be divided into three categories: subprograms for time-dependent tasks, subprograms for processing messages that have been received, and subprograms for executing the processes. The microcontroller's built-in watchdog timer receives the strobe signal while the infinite loop is being performed. This guarantees that, in the event that the microcontroller software is blocked, the watchdog timer will reset the microcontroller. If necessary, the main program's execution is halted so that the proper interrupt subprograms can be run. At least, there are serial interrupt subprograms and timer interrupt subprograms.

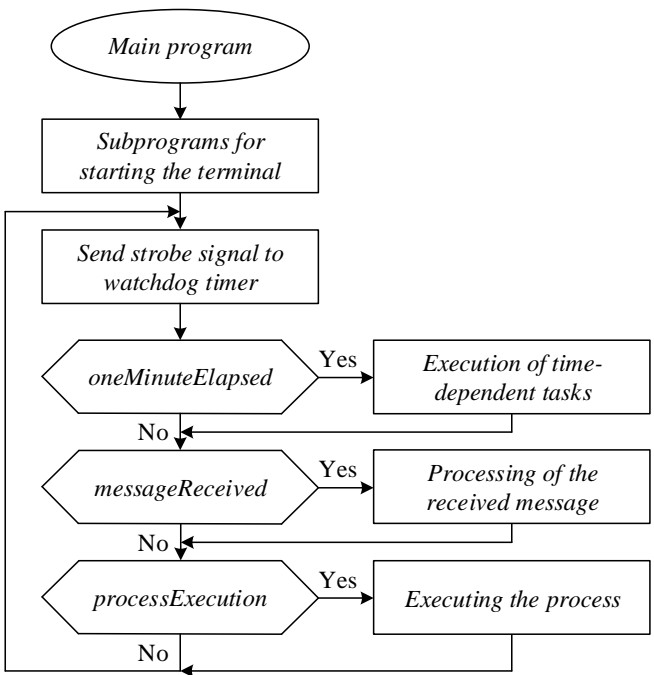

**Figure 7.** Main program.

The Initialization of global variables, Configuration of other components, Configuration of the microcontroller, and Recovering from a power outage subprograms are all included in the group of subprograms for starting the terminal. All the important microcontroller registers that determine timer operation permission, timer operation mode, timer trigger frequency, UART operation permission, UART operation mode, interrupt permission, interrupt priorities, etc., are configured by the subprogram Configuration of the microcontroller. All global variables are given their initial values during the subprogram Initialization of global variables. This is required because, after the microcontroller is powered on, these variables, which are stored in the SRAM of the device, have some arbitrary values. All other terminal components, save the microcontroller, are configured by the subprogram Configuration of other components. Setting the display and/or LEDs, configuring the ports, reading the real-time clock, and initializing the 5G/4G/3G/GPRS/GSM modem by turning on the flag activatedModemInitialization are all included in this. Modem initialization is followed by the initialization of key variables for modem operation (proper setting of the state and status variables, initialization of additional control variables, initialization of the first state's control variables, etc.), as well as the start of the process' first state's activity. The subprogram Recovery from a power outage assures that the terminal will continue to operate after the power is restored and that the global variables will be read from a non-volatile (permanent) memory.

The subprogram Execution of time-dependent tasks is at least one of the members of the group of subprograms for time-dependent tasks. After setting the flag oneMinuteElapsed, the main program calls this subprogram. The time is read in that function, and the required tasks are begun at the right times to be executed. Each time the appropriate time interval has passed, the flag oneMinuteElapsed is enabled in the interrupt subprogram of the relevant timer. A set of actions conducted in response to an event is represented by the subprograms used to execute the processes. The process transitions through various states as it is being executed. The process of making decisions in demand side management [31] and the process of controlling electric car chargers optimally to reduce network load [8], employing various tactics [9–13] depending on the tariff, are examples of more complex processes.

The security of IoT devices [32] and terminals [33] presents a unique set of issues that must be addressed. These issues are based on numerous strategic and tactical suggestions as well as best practices in successful and secure implementations [34,35].

## 8. Drivers

The light terminal also requires the relevant drivers to be implemented in order for the software described in the previous section to operate correctly. There are two viable approaches to implementing the conventional terminal: using the aforementioned drivers or a bespoke operating system (FreeRTOS). Operating systems such as Linux or Windows are used to run the extended terminal.

The following interrupt subprograms, the serial interrupt subprogram and the timer interrupt subprogram, are the key drivers. The timer interrupt subprogram is monitoring the expiration of the one-minute period, controlling the keys and processing the appropriate number of timers required for the operation of the entire microcontroller software. If extra serial ports are built using external integrated circuits, serial communication (with IoT network infrastructure, an owner, or a cloud) is carried out in this subprogram. The appropriate time interval between the two interrupts is recorded in the timer registers during each execution of this subprogram, ensuring that the right number of times per second are called into this procedure. The oneMinuteElapsed flag is activated by the timer interrupt subprogram each time a minute has passed.

To determine when a set amount of time has passed, a timer is used. The timer interrupt subprogram processes the timer by decrementing its value and taking the required action when it reaches zero. Figure 8 depicts this timer operation algorithm. Timers are crucial in ensuring that the execution of microcontroller programs is not stopped while they wait for a relevant event to occur in the external environment. The microcontroller is capable of supporting multiple independent timer interrupts with various interrupt durations.

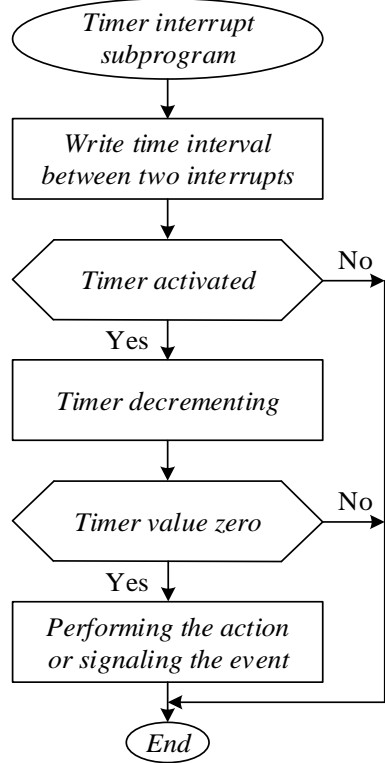

**Figure 8.** Timer interrupt subprogram.

The serial interrupt subprogram is in charge of receiving and delivering messages via the UART microcontroller in serial communication. When one byte of the message has been

received or delivered and the next byte of the message needs to be sent, the serial interrupt subprogram is invoked. The serial interrupt service procedure also recognizes when a message has finished being received. As previously mentioned, a GSM/GPRS modem is connected serially to the microcontroller using its UART, and it is through this serial connection that the microcontroller realizes three different types of communications: communication with the modem itself, interaction with the cloud, and communication with the service technician. The messages in these communications are, therefore, received and sent by the serial interrupt service procedure. Through a serial connection to a connector (discussed in Section 5) using the microcontroller's UART, a service technician can be reached by the microcontroller. The AT protocol, or, more specifically, the AT commands, are used by the microcontroller to connect with the modem. The 5G/4G/3G/GPRS/GSM data connection already in place uses the cloud communication protocol to enable communication between the microcontroller and the cloud. Either directly through a cable connection via the connector or indirectly through a wireless connection via a modem within the established 5G/4G/3G/GPRS/GSM data connection, the microcontroller communicates with the repairman.

It is, therefore, important to include a variable for storage of information about the current serial communication mode, i.e., the variable serialCommunicationMode, due to the various packet structures used in serial communication. The AT COMMAND constant's standard value for this variable indicates that the AT protocol's packet structure is being followed by the present communication's packet structure. The only time this variable deviates from its default value is when communicating with the cloud, which occurs from the time a 5G/4G/3G/GPRS/GSM data connection is established to the time it is terminated, at which point its value is equal to the CLOUD DATA constant. Figure 9 illustrates the algorithm of the serial interrupt service function. The serial interrupt subprogram checks the serialCommunicationMode variable and executes one of the following subprograms based on its value: AT communication or cloud communication.

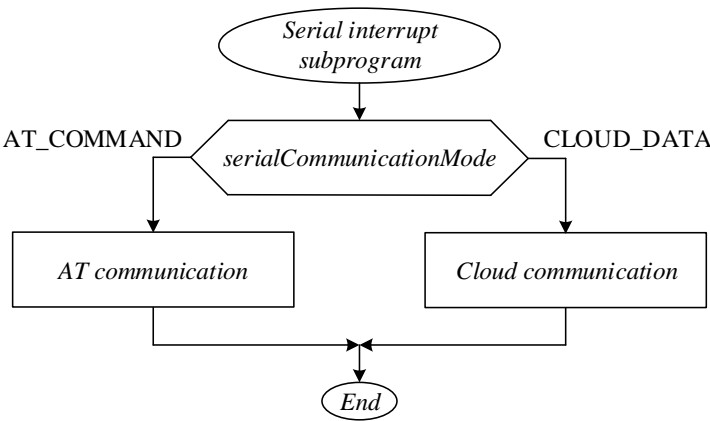

**Figure 9.** Serial interrupt subprogram.

## 9. The Subprograms for Executing the Processes

In order to more clearly explain the subprograms from this group, it is worth first explaining the process execution mechanism itself. A process is a closed set of activities undertaken in response to an event to generate an output. During execution, a process goes through various states. Process state changes are conditioned by corresponding events. In each state, appropriate action is taken.

The method of realization of the process is as follows. Each process is assigned: activation flag, state variable, status variable, startup subprogram, execution subprogram, state subprogram, process control variables, and state control variables.

The process can be found in different states. Information about the state of the process is stored in the corresponding state variable.

Corresponding events cause the state of the process to change. Information about these events is stored in appropriate flags or variables.

The process is executed in several cycles. In one cycle of the process, the activity of the state in which the process is at that moment is executed once.

In each state of the process, a corresponding activity is executed. The number of different activities is equal to the number of different process states, so one activity corresponds to each process state.

Repetition of the same activity occurs in the case of keeping the process in the corresponding state for more than one cycle.

The process is executed within the appropriate execution subprogram, which is executed in the main program each time it passes through the infinite loop as long as the activation flag is active.

One cycle of the process corresponds to one pass through the infinite loop of the main program.

Depending on the value of the state variable, the execution subprogram calls the corresponding state subprogram.

Within the status subprogram, the appropriate activity is executed, checking whether the appropriate events have occurred and, depending on that, the process remaining in the same state or the appropriate change in the process state.

Starting the process, that is, changing the state of the process from the state of inactivity to the first state of execution, is completed by calling and executing the start subprogram. As part of the startup subprogram, the corresponding activation flag is activated, the state variable and status variable are set appropriately, other process control variables are initialized, the control variables of the first state are initialized, and the activity of the first state of the process is started.

Changing the state of the process from the current state to the next state is completed by setting the state variable accordingly, initializing the control variables of the next state, and starting the next state activity.

Stopping the process, that is, changing the state of the process from the current state to the state of inactivity, is completed by deactivating the activation flag, setting the state variable and the status variable accordingly, and, if necessary, setting the appropriate parameters that affect the next start of the process, the start of other subprograms or processes, etc. If necessary, stopping the process can be completed within a separate subprogram, i.e., within the stop subprogram.

Putting the process into the waiting state, i.e., changing the process state from the current state of execution to the state of waiting, is completed by deactivating the activation flag, setting the state variable accordingly, and, if necessary, starting the detection of appropriate events that will return the process from the state of waiting to the state of execution.

The status of the process, that is, the information whether the process has successfully generated a result, is stored in the corresponding status variable. In the same variable, information about the reason for the failure of the process is stored in case of unsuccessful completion of the process.

This description of the process is a model-driven development, i.e., a model for object-oriented programming. The advantage of this way of programming is that it enables much faster and better-quality development and maintenance of software. On the other hand, the disadvantage is that the programmer has less control over the speed of execution and the amount of program and working memory required for the operation of the resulting software. This can be limiting for embedded software applications. However, in the following text, we will describe how the process model thus obtained is translated into program code in such a way as to achieve all the advantages and neutralize the mentioned disadvantages on the example of complex subprograms.

In the further text of this chapter, a detailed explanation of one of the more complex subprograms from this group follows, that is, the subprogram of creating a complete daily (Z) report. The explanation is provided for the fiscal cash register CR401 manufactured by

Intracom, and the same can be applied to other fiscal cash registers. The purpose of the CreateCompleteZReport subprogram is to execute the process of creating a complete Z report. Figure 10 shows a state diagram of the complete Z report workflow.

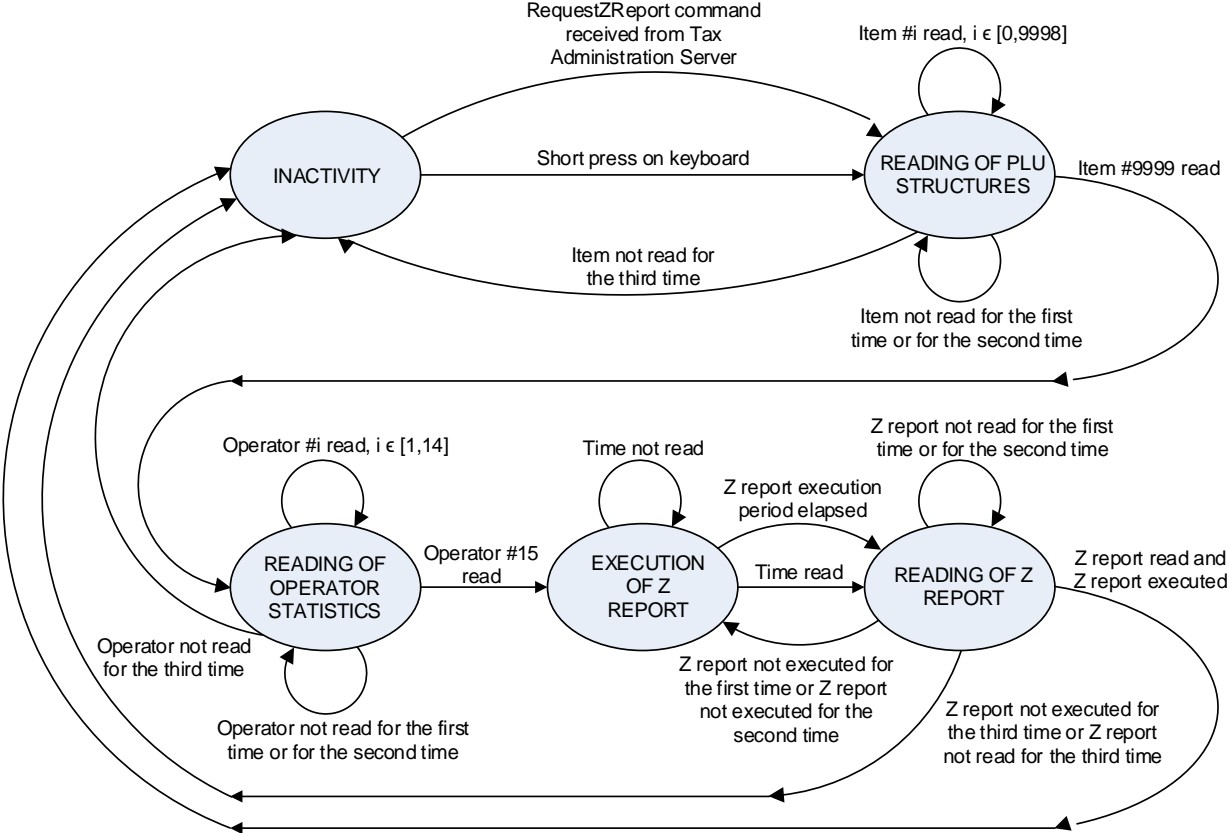

**Figure 10.** A state diagram of the complete Z report workflow.

The result of the process, that is, the purpose of the process of creating a complete Z report, is to prepare a daily turnover report by items, a daily turnover report by operators (cashiers), and a daily report for sending to the PO server, that is, to ensure the reading of data from the fiscal cash register, the formatting of such read data into reports, and the placement of such generated reports into the terminal's memory. The PO server is the taxpayer's server, which allows a taxpayer to monitor all relevant data and possibly adjust the parameters available to that taxpayer.

The process can be found in five states. The names of the states are determined in accordance with the procedure that is carried out in that state in the fiscal cash register. The following is an overview of the status and corresponding activities, events, and status changes.

State of inactivity. The process is initially in this state. From this state, the process can move to the state of reading the PLU (Price Look-Up) structure of the article if a key activation of less than 5 s occurs or a RequestZReport command is received from the PO server.

Reading of PLU article structures state. The purpose of this state is to prepare a daily turnover report by item to be sent to the PO server. Therefore, the activity of the process in this state is reading the PLU structure of items from the fiscal register, formatting the thus obtained data, and placing it in the memory of the terminal in the data block seriesKZReports in the PLUReport field. In the regular execution of the process, this activity is executed 10,000 times because the fiscal cash register CR401 manufactured by Intracom has the ability to work with 10,000 items. The process remains in the same state as long as the PLU structure of the item with sequence number from the set of numbers

0 to 9998 happens to be read. When the PLU structure of the item with sequence number 9999 happens to be read, the process goes into the state of reading of operator statistics. The process remains in the same state until it happens that some PLU structure is not read in the first or second attempt. If it happens that some PLU structure is not read in the third attempt, the process transitions into the inactive state and places the information about the unsuccessful execution of the process because the fiscal cash register does not respond to the command to read the PLU structure in the status variable of statusZReport.

Reading of operator statistics state. The purpose of this state is to prepare a daily turnover report by cashiers. Therefore, the activity process in this state is the reading of the operator's statistical data (hereinafter referred to as the operator structure) from the fiscal cash register, formatting the thus obtained data, and placing it in the terminal's memory in the data block seriesKZReports in the operatorReport field. In the regular execution of the process, this activity is executed 15 times because the fiscal cash register CR401 manufactured by Intracom has the ability to work with 15 cashiers. The process remains in the same state as long as the operator structure with the sequence number from the set of numbers 1 to 14 happens to be read. When the operator structure with the sequence number 15 happens to be read, the process transitions into the execution of Z report state. The process remains in the same state until it happens that some operator structure is not read in the first or second attempt. If it happens that some operator structure is not read in the third attempt, the process transitions into an inactive state and places information about the failed execution of the process because the fiscal cash register does not respond to the command to read operator statistics in the status variable statusZReport.

Execution of Z report state. The purpose of this state is the creation of a daily report, that is, the execution of the Z report in the fiscal cash register in the stipulated time period. Therefore, the activity of the process in this state is the execution of the Z report in the fiscal cash register, the detection of the event that the Z report execution procedure in the fiscal cash register has been completed, and the initiation of the event detection procedure that the scheduled time period for the execution of the Z report has expired. Detection of the event that the Z report execution procedure in the fiscal cash register has been completed is conducted by asking whether the fiscal cash register responds to the time reading command (time reading was chosen as one of the simpler commands). Such detection is necessary for the following reasons: the fiscal cash register does not respond to the command to execute the Z report, the fiscal cash register does not respond to any command during the execution of the Z report, and the time required by the fiscal cash register to execute the Z report is not constant but depends on the number of items sold in that day and other factors. Initiation of the event detection procedure that the scheduled time period for execution of the Z report has expired is completed by activating the appropriate timer. In regular process execution, this activity is executed once. The time reading is performed several times, which depends on the ratio of the time required by the fiscal cash register to execute the Z report and the time waiting for the response from the fiscal cash register. If the event occurs that the time has not been read, the process remains in the same state. If the event occurs that the time has been read (the event that the Z report execution procedure in the fiscal cash register is completed) or that the scheduled time period for the Z report execution has expired, the process switches to the reading of Z report state.

Reading of Z report state. The purpose of this state is to prepare a daily report for sending to the PO server. Therefore, the activity of the process in this state is the reading of the last Z data (hereinafter the last Z report) from the fiscal cash register, formatting the thus obtained data, placing it in the memory of the terminal in the data block seriesKZReports in the field descriptionZReport, and detecting the event that the Z report execution procedure is successfully completed in the fiscal cash register. This detection is performed by checking whether the number of the last Z report read from the fiscal cash register is different from the number of the last Z report stored in the terminal. In regular process execution, this activity is executed once. If events occur that the last Z report was read and that the Z report was successfully executed, the process transitions into the inactive state and places

the information about the successful execution of the process in the status variable status. The process remains in the same state until it happens that the last Z report is not read in the first or second attempt. If it happens that the last Z report is not read in the third attempt, the process transitions into the inactive state and places the information about the unsuccessful execution of the process because the fiscal cash register does not respond to the command to read the last Z report in the status variable of statusZReport. If the events occur that the last Z report was read and the Z report failed to execute on the first or second attempt, the process enters the execution of Z report state. If it happens that the last Z report was read and the Z report was executed unsuccessfully in the third attempt, the process transitions into the inactive state and places the information about the failed execution of the process because the fiscal cash register does not execute the Z report in the status variable of statusZReport.

The flag of the activation of this process is the workCompleteZReport flag. The execution subprogram of this process is the CreateCompleteZReport subprogram. The state variable of this process is the status variable, that is, the same variable that is used to store information about the busy status of communication with the fiscal cash register. The status variable of this process is the variable statusZReport within the field descriptionZReport of the pointed member of the block stringKZReport in the terminal memory. The process startup subprogram is the InitializationWorkCompleteZReport subprogram.

The process is started by the InitializationRunCompleteZReport within the timer 0 interrupt subprogram or the ModemMessageProcessing subprogram or the PowerLossRecovery subprogram. The process is executed in the CreateCompleteZReport subprogram.

The CreateCompleteZReport subprogram is executed in the main program as long as the workCompleteZReport flag is active. All the time during the process of creating a complete Z report, information about the state of the process is stored in the status variable. For all time during and after the process of creating a complete Z report, the status information is stored in the variable statusZReport within the field descriptionZReport of the pointed member of the block stringKZReport in the terminal memory.

Figure 11 shows the algorithm of the CreateCompleteZReport subprogram. The main purpose of the CreateCompleteZReport subprogram is to ensure the execution of the process of creating a complete Z report, that is, the execution of the corresponding activity, that is, the corresponding state subprogram, depending on the state of the process.

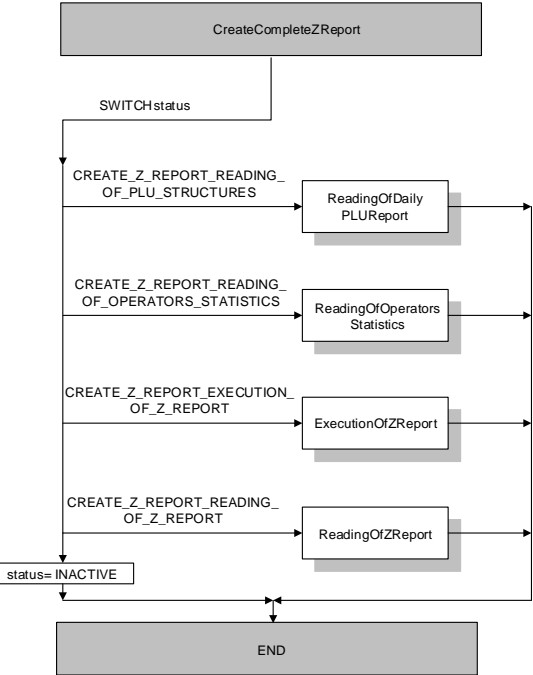

**Figure 11.** CreateCompletZReport subprogram algorithm.

Therefore, at the beginning of the CreateCompleteZReport subprogram, the status of the process is checked, that is, the status variable is checked. Depending on the state of the process, the corresponding previously described activity is executed by calling and executing the corresponding state subprogram.

If the process is in the state of reading of PLU structures, the ReadingOfDailyPLUReport subprogram is called. If the process is in the state of reading of operator statistics, the ReadingOfOperatorStatistics subprogram is called. If the process is in execution of Z report state, the ExecutionOfZReport subprogram is called. If the process is in the state of reading of Z report, the subprogram ReadingOfZReport is called. If the process is not in any of the mentioned five states, the variable status is set to the value of the constant INACTIVE.

## 10. Results and Discussion

Table 1 shows the results of applying the described model-driven development method to the terminal for remote control of fiscal cash registers and a comparison with the results of the development of a terminal for remote control of smart meters without applying this method. The basic software of this terminal included the preparation of appropriate reports and communication with the taxpayer's server (PO Server), while, in the advanced version, the software was upgraded to support the preparation of appropriate reports and communication with the tax administration control center. We can see that it took 30 days to develop the basic software of the terminal. The same period was required to upgrade the software to the advanced variant, while the maintenance lasted about 60 days. During the maintenance period, the release of 24 software versions occurred, which means that the average response time in that period was 2.5 days, all in total for software correction (bug correction or functionality refinement) and testing. Compared to the terminal for remote control of smart electricity meters, which was developed without using the model-driven development method, all the mentioned parameters were at least two times higher.

**Table 1.** The results of applying model-driven development method to the terminal for remote control of fiscal cash registers and a comparison with the results of the development of a terminal for remote control of smart meters without applying this method.

| Activity | Terminal for FCR | | Terminal for Smart Meter |
|---|---|---|---|
| | **Duration Period** | **Number of Versions** | **Duration Period** |
| Development of basic variant | 30 days | N/A | >60 days |
| Development of advanced variant | 30 days | N/A | >60 days |
| Maintenance | 60 days | 24 | >120 days |

Tables 2 and 3 show more details about all 24 software versions in the maintenance period. Table 2 lists the duration and activity performed for each software version. Activities performed during the maintenance period are bug fix, new feature adding, and functionality improvement. The longest duration of the development of one version of the software was 10 days, and the shortest was 1 day. Table 3 lists the total number of software versions produced and the total time spent on each of the listed activities. Further, 45% of the time was spent on improving functionalities, 35% of time was spent on fixing bugs, and 20% of time was spent on adding new features. Therefore, the most time was spent on functionality improvement, while the least time was spent on adding new features. This shows that, no matter how much time is spent on the precise description of the product, that is, defining the technical requirements as precisely as possible, some improvements are always observed only after the realization of the product.

**Table 2.** The results of applying model-driven development method to the terminal for remote control of fiscal cash registers during the maintenance period for each version of the software.

| Terminal for FCR | | |
|---|---|---|
| **Maintance Version** | **Duration Period (Days)** | **Activity** |
| Version 1 | 1 | Functionality improvement |
| Version 2 | 4 | Bug fix |
| Version 3 | 7 | Bug fix |
| Version 4 | 8 | New feature adding |
| Version 5 | 1 | Functionality improvement |
| Version 6 | 10 | Functionality improvement |
| Version 7 | 2 | New feature adding |
| Version 8 | 5 | Functionality improvement |
| Version 9 | 2 | Bug fix |
| Version 10 | 1 | Functionality improvement |
| Version 11 | 1 | Bug fix |
| Version 12 | 1 | Functionality improvement |
| Version 13 | 2 | Functionality improvement |
| Version 14 | 1 | Bug fix |
| Version 15 | 3 | Functionality improvement |
| Version 16 | 1 | Functionality improvement |
| Version 17 | 1 | Bug fix |
| Version 18 | 1 | Bug fix |
| Version 19 | 1 | New feature adding |
| Version 20 | 1 | Bug fix |
| Version 21 | 3 | Bug fix |
| Version 22 | 1 | Functionality improvement |
| Version 23 | 1 | New feature adding |
| Version 24 | 1 | Functionality improvement |

**Table 3.** The results of applying model-driven development method to the terminal for remote control of fiscal cash registers in the maintenance period in total.

| Terminal for FCR | | | |
|---|---|---|---|
| **Activity** | **Software Versions Total** | **Duration Period Total** | |
| | **Number** | **Days** | **Percentage** |
| Functionality improvement | 11 | 27 | 45% |
| Bug fix | 9 | 21 | 35% |
| New feature adding | 4 | 12 | 20% |

These experimental results clearly confirm the importance of applying the model-driven development method in software development. This example highlights the main feature of this method, which is reflected in the fact that it significantly reduces the time required for subsequent software changes.

Table 4 shows the simulation, that is, the prediction of the time required for the development and maintenance of the terminal for remote control of renewable-energy-powered EV charging station, using the model-driven development method. The initial assumption is that the development and maintenance of each individual terminal from the terminals for remote control of the smart battery, the smart meter, and the EV charger do not require more time than the time spent on the development and maintenance of the terminal for the remote control of fiscal cash registers. Furthermore, it was taken into account that the terminal for remote control of the station combines the functionalities of these four terminals. Based on this assumption and the stated fact, it was concluded that the development of the terminal for remote control of the station will not take more time than four times the time required for the development and maintenance of the terminal for remote control of fiscal cash register.

**Table 4.** The simulation of applying model-driven development method to the terminal for remote control of the station.

|  | Terminal for FCR | Terminal for Station |
|---|---|---|
| **Activity** | **Duration Period (Days)** | **Duration Period (Days)** |
| Development of basic variant | 30 days | <120 |
| Developmemt of advanced variant | 30 | <120 |
| Maintance | 60 | <240 |

The contributions of this paper are:

- In comparison with other works, none of the previously mentioned approaches offer such a system, that is, a terminal with unified control of chargers, renewable sources, smart battery, smart meter, and fiscal cash register, as well as categorization of such a terminal.
- Furthermore, in addition to the realization of the terminal hardware that enables communication with all these devices, this paper also provides a design development model, that is, a model for both structured and object-oriented implementation of embedded software, which can be used for application in smart cities systems, smart metering, smart home, smart grid, and smart energy management in general.
- This design development model provided excellent results in practice when applied to terminals for remote control of fiscal cash registers, enabling at the same time much faster and better software development and maintenance while maintaining a sufficient level of control over execution speed and the amount of required working and program memory.

## 11. Conclusions

This article has considered the design of a terminal for remote control of a station powered by renewable energy sources for charging electric vehicles. After presenting the architecture of the system for station remote control, a block diagram of the terminal hardware was described. It was shown that the central part of that system should be one device—a terminal with unified control of chargers, smart meter and fiscal cash register, smart battery, and renewable sources. This unique terminal can replace four separate terminals, which separately provide remote control of the mentioned devices. It has been determined that a microcontroller with the proper memory, communication ports, keys, visual indications, a real-time clock, and a power adapter can serve as the basis for the terminal hardware realization. There are three suggested hardware implementation alternatives, together with the necessary drivers. The operation of the terminal software is based on initializing all necessary registers, variables, parts, and processes, followed by running the program's endless main loop, which executes all essential operations and is sporadically stopped by interrupt subprograms. An overview of the complete terminal software is provided. The implementation of very complex subprograms for process execution is also explained in detail. The model-driven development was used, which enables the application of object-oriented programming. This is a very advanced method for embedded software. Potential inconveniences or problems of this method could be a lack of memory. From the other side, this method enables much faster and better-quality development and maintenance of software while maintaining control over the speed of execution and consumption of program and working memory. The practical application of this model during the implementation of terminals for remote control of fiscal cash registers demonstrated satisfactory results. In further work, the development of a compiler that would automatically generate program code based on the model can be considered. Moreover, the analysis of which renewable sources can be used at commercial and industrial charging stations, as well as the analysis of whether only vehicle batteries can be used instead of a special independent smart battery, can be further research directions.

**Author Contributions:** Conceptualization, J.V., G.S. and M.P.; methodology, J.V., G.S. and M.P.; software, J.V.; validation, J.V.; formal analysis, J.V., G.S. and M.P.; investigation, J.V., G.S. and M.P.; resources, J.V., G.S. and M.P.; data curation, J.V.; writing—original draft preparation, J.V. and G.S.; writing—review and editing, J.V., G.S. and M.P.; visualization, J.V.; supervision, M.P.; project administration, J.V. and M.P.; funding acquisition, J.V. All authors have read and agreed to the published version of the manuscript.

**Funding:** This research received no external funding.

**Data Availability Statement:** Data available on request.

**Conflicts of Interest:** The authors declare no conflict of interest.

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
