# Peer review of "Model-Driven Developed Terminal for Remote Control of Charging Station for Electric Vehicles Powered by Renewable Energy"

_electronics, doi:10.3390/electronics12081769_

Round 1

Reviewer 1 Report

 Terminal for remote control of renewable energy sources powered station for electric vehicles charging has been presented in this paper. The topic is interesting and need of time, however I have following concerns in the paper; 

  • The introduction section is too short to understand the background and motivation of the paper. Please expand the introduction section by adding more details.  
  • Section 9 should be after introduction section, please change the position of the section related work.  
  • Some typos and grammatical mistakes should be fixed before further proceeding. Some sentences are too long to understand.  
  • Some simulation-based analysis should be presented to validate the proposed model. Please add some simulation-based case study. 
  • Quality of the figure 2 can be improved.   
  • The result tables can be added for the readers easiness. Tables such as the already worked survey etc.  
  • The role of the prosumer in the context of the electricity market is not added. 
  • The references are required in different sections such as in the second paragraph of section 4.  
  • In section 8, how to check the validation of the whole process? what criteria have been fixed to show the completion of the process?  

Thanks, and Regards

Author Response

We appreciate the time and efforts of Reviewer #1 to review the manuscript and we want to thank Reviewer #1 for comments we found very useful for the improvement of our manuscript.

Reviewer 2 Report

Dear Authors,

The paper presents interesting aspects about charging stations for electric vehicles.

Still several aspects should be solved related to the “renewable” point of view:

1.       A description in the first part of the paper is required related to what the authors understand under “Renewable Energy Sources Powered Station for Electric Vehicles Charging”, because installing for example PV systems on a charging station might contribute only a few % to the energy required for the vehicles.

2.       L28: In order to discuss about the “fully benefits of electric vehicles” I recommend also discussing about the problems related to the production and recycling processes required for the batteries of such vehicles. Otherwise the problems are just transferred from highly industrialized countries to lower ones.

3.       L86 : a description of what a “smart battery” is, is firstly required.

4.       Also using an extra battery to the one used in vehicles might not be an economically feasible solution. What is the lifetime of such a battery continuously charged and discharged?

5.       Chapter 9, Related Work, should be included in the Introduction of the paper.

6.       Please add to the conclusion section also the inconveniences, problems of the new proposed system.

The paper has 37% similarity using Turnitin with other papers of the authors. Please resolve, in this form it is not publishable!

I hope that my comments will be useful.

Sincerely,

Reviewer

Author Response

We appreciate the time and efforts of Reviewer #2 to review the manuscript and we want to thank Reviewer #2 for comments we found very useful for the improvement of our manuscript.

Reviewer 3 Report

The topic presented in the paper is valid and still on time. Taking into account the recommendations of many governments on the world and many regulations finding new sources of vehicle fuels is very important. In the reviewed paper, the Authors presented the terminal for remote control of renewable energy sources powered station for electric vehicles charging has been presented in this paper. This terminal enables remote control of electric vehicle chargers, smart storage batteries, smart electricity meters, and cash registers, as well as, remote control of renewable energy sources and other devices within the station. This terminal also makes stations for electric vehicle charging powered by renewable energy sources more accessible to electric vehicle users, to electricity distribution system operators, to electricity supplier operators, to tax administration operators, and finally to users and owners of the station. Therefore, communication and control with all these devices and systems is integrated in one device. So, the topic is worth to recognition. In my opinion, the paper can be published, after taking into account the following remarks:

- in the abstract section, the Authors described what they did and present in the paper. It looks like a short review of the paper's content. At the end of the abstract section, there is a lack of short information about the paper's results and future usefulness,

- the introduction to the subject of the article presented in the Introduction section is very short and should be extended,

- in the Introduction section, the Authors presented a short background of the topic mentioning, among others, the increase in production and use of electric vehicles, the air pollution reduction problems, stations for electric vehicle charging, etc. It is very good, but the Authors should also add some characteristics of electric vehicles because the article is dedicated to them. Thus, the Authors should refer to the latest positions in the scientific literature in this field, i.e. "Electric vehicles - problems and issues" doi 10.1007/978-3-030-35543-2_14; "Thermal Performance Optimization of Multiple Circuits Cooling System for Fuel Cell Vehicle" doi.org/10.3390/su15043132  One short paragraph in the Introduction section will be enough,

- the Authors used in the paper text some acronyms without explanation at the same time. The Authors are asked to check in the whole text the acronyms and add missing explanations,

- there is a lack of a "Literature review" section dedicated to previous research works on the presented in the paper topic. It would be good if the Authors consider adding such a section,

- according to the information presented by the Authors: ..."In Figure 1 block diagram of the architecture of the system for remote control of renewable energy sources powered station for electric vehicles charging [2] is shown."... the figure 1 has a source in [2]. Do the Authors have written permission for further usage of this figure? Such written permission is usually obligatory for the journal office,

- did the Authors calculated the cost of the proposed driven developed terminal for remote control of renewable energy sources powered station for electric vehicles charging? The cost of new solutions usually plays an important role in the successful idea implementation,

- on the figure called "Figure 8. Timer interrupt service routine", what does it mean the empty circle at the end of the scheme? It should by "end" probably inside,

- the Authors are asking to check the whole paper text because some typos inside can be found,

- the idea proposed in the article is interesting, but the detailed technical requirements of the proposed solutions are missing,

- it is recommended to replace the name of the section called "9. Related Work" to "9. Discussion",

- is like follows "10. Conclusion". Should be like follows "10. Conclusions",

-  the Conclusions section is written in a very general way and should be extended by adding some detailed conclusions from presented in the paper analysis. 

Author Response

We appreciate the time and efforts of Reviewer #3 to review the manuscript and we want to thank Reviewer #3 for comments we found very useful for the improvement of our manuscript.

Reviewer 4 Report

1. The contribution of the paper need to be specified. Put it in the bullet form.

2. Title mentioned "Remote Control of Renewable Energy Sources", add more discussion about renewable energy resources for the EV application.

3. What is the importance of remote control. What about the cyber thread in such applications? Add a discussion about it. 

4. Section 3.3: again, what about cyber security? 

5. Add some real time examples of the system described with proper references. 

6. What about the economics of the remote control?

7. Add some simulation results of such cases.

8. Update the references and use the latest one from 2021, 2022.

Author Response

We appreciate the time and efforts of Reviewer #4 to review the manuscript and we want to thank Reviewer #4 for comments we found very useful for the improvement of our manuscript.

Reviewer 5 Report

In this paper, a terminal for remote control of a renewable energy power plant for charging electric vehicles was presented.

This research is considered to be a very interesting field as a research field that fits the recent technological trend.

However, there are some areas that must be improved.

1. The title of the thesis shows that remote control is performed to efficiently charge electric vehicles using renewable energy.

If so, I think that it should be described in which scenario the remote control was performed in detail and the results obtained efficiently and cost-effectively.

There seems to be a difference between the title of this study and the actual contents.

2. As I said before, the subject of this study is great. However, it seems somewhat lacking in academic content and originality to support this.

Although the authors have divided many sections into detailed descriptions, it is considered to be a list of technical reports.

3. Please ask how to change the composition (order) of the paper.

It would be good to include the related research section of Chapter 9 in the introduction of Chapter 1 and mention the contents of previous studies and the necessity and originality of this study.

I think it will be of great help to readers who are interested in this study.

4. The practical application of this model in practice during the implementation of terminals for remote control of fiscal cash registers gave satisfactory results.

: What satisfactory results did you get?

It would be a qualitatively improved thesis if the results obtained through the implementation of this study were modified to stand out by preparing a table or characteristic graph separately to see what results were obtained in detail.

Author Response

We appreciate the time and efforts of Reviewer #5 to review the manuscript and we want to thank Reviewer #5 for comments we found very useful for the improvement of our manuscript.

Round 2

Reviewer 1 Report

well addressed 

Author Response

We appreciate the time and efforts of Reviewer #1 to review the
manuscript and we want to thank Reviewer #1 for comments and suggestions
we found very useful for the improvement of our manuscript

Reviewer 2 Report

Dear Authors,

The paper was improved, but I recommend adding more experimental data to it.

At least a detailed simulation of the proposed solution is required.

Otherwise it just presents solutions without the proof that they are actually working.

I hope that my comments will be useful.

Sincerely,

Reviewer

Author Response

We appreciate the time and efforts of Reviewer #2 to review the
manuscript and we want to thank Reviewer #2 for comments and suggestions
we found very useful for the improvement of our manuscript.

Given that the other four reviewers accepted the paper after the first round of revision, we hope that the work is now acceptable to you as well, and therefore ready for publication.

Reviewer 4 Report

no more comments

Good work.

Author Response

We appreciate the time and efforts of Reviewer #4 to review the
manuscript and we want to thank Reviewer #4 for comments and suggestions
we found very useful for the improvement of our manuscript

Reviewer 5 Report

Thank you for your reply

Author Response

We appreciate the time and efforts of Reviewer #5 to review the
manuscript and we want to thank Reviewer #5 for comments and suggestions
we found very useful for the improvement of our manuscript